# A Survey of Tick Infestation and Tick-Borne Piroplasm Infection of Cattle in Oudalan and Séno Provinces, Northern Burkina Faso

**DOI:** 10.3390/pathogens11010031

**Published:** 2021-12-28

**Authors:** Paul Franck Adjou Moumouni, Germaine Lim-Bamba Minoungou, Christian Enonkpon Dovonou, Eloiza May Galon, Artemis Efstratiou, Maria Agnes Tumwebaze, Benedicto Byamukama, Patrick Vudriko, Rika Umemiya-Shirafuji, Hiroshi Suzuki, Xuenan Xuan

**Affiliations:** 1National Research Center for Protozoan Diseases, Obihiro University of Agriculture and Veterinary Medicine, Obihiro 080-8555, Hokkaido, Japan; chakirou82@yahoo.fr (P.F.A.M.); eloizagalon@gmail.com (E.M.G.); aefstratiou@outlook.com (A.E.); tumwebazeaggie@gmail.com (M.A.T.); benards.benedicto4@gmail.com (B.B.); vpato2009@gmail.com (P.V.); umemiya@obihiro.ac.jp (R.U.-S.); hisuzuki@obihiro.ac.jp (H.S.); 2Laboratoire National d’Elevage, Rue du 11 Décembre, Ouagadougou 09 BP 907, Burkina Faso; minoungou.germaine@gmail.com; 3Dierenartsen Zonder Grenzen–Vétérinaires Sans Frontières Belgique, Burkina Faso Office, Avenue Charles de Gaulle, Ouagadougou 06 BP 9508, Burkina Faso; c.dovonou@vsf-belgium.org

**Keywords:** *Babesia*, Burkina Faso, cattle, epidemiology, *Theileria*, tick species

## Abstract

In this study, cattle farms located in Oudalan and Séno, two provinces in the Sahel region, northern Burkina Faso, were surveyed. Cattle owners were interviewed, cattle were examined for tick infestation, and ticks as well as blood samples were collected during the dry season (October). Blood DNA samples were tested for *Babesia* and *Theileria* infections using nested PCRs and sequencing. A total of 22 herds, 174 Zebu cattle were investigated at 6 different sites. Overall, 76 cattle (43.7 %) from 18 farms (81.8%) were found infested with ticks. Cattle in Séno, adult cattle (>5 years) and those owned by the Fulani ethnic group were significantly (*p* < 0.05) more likely to be tick-infested. A total of 144 adult ticks belonging to five species namely: *Hyalomma impeltatum*, *Hyalomma impressum*, *Hyalomma rufipes*, *Rhipicephalus evertsi evertsi*, and *Rhipicephalus guilhoni* were collected from the animals. Piroplasms were detected in the blood DNA of 23 (13.2%) cattle. The cattle in Séno and adult cattle were significantly more likely to be piroplasm-positive. Five pathogens diversely distributed were identified. *Theileria mutans* (12/174), *Babesia bigemina* (5/174), *Theileria annulata* (3/174), and *Theileria velifera* (3/174) were detected for the first time in northern Burkina Faso, whereas *Babesia occultans* (1/174) was found for the first time in cattle in West Africa. The analysis of the sequences, including *B. bigemina RAP-1a*, *T. annulata Tams1* genes, and the 18S rRNA genes of all the five protozoa, revealed identities ranging from 98.4 to 100% with previously published sequences. Phylogenetic analysis based on the 18S rRNA gene sequences located north Burkina Faso piroplasms in the same clade as isolates from Africa and other regions of the world. Notably, *T. mutans* sequences were distributed in two clades: the *T. mutans* Intona strain clade and the *Theileria* sp. (strain MSD)/ *Theileria* sp. B15a clade, suggesting the presence of at least two strains in the area. These findings indicate that the control of ticks and tick-borne diseases should be taken into account in strategies to improve animal health in the Sahel region.

## 1. Introduction 

Ticks are hematophagous ectoparasites and are considered the most important vectors of disease-causing pathogens in domestic and wild animals [1]. Cattle are particularly affected by tick and tick–borne diseases (TTBD), with around 80% of the world’s population at risk and global losses estimated to be 22–30 billion USD per year [2]. In sub-Saharan Africa (SSA), the major part of these losses is attributed to piroplasmosis [3,4,5] caused by hemoprotozoan parasites of the order Piroplasmida, genera *Babesia* and *Theileria*. Piroplasms and ticks have specific interactions and many piroplasms are associated with particular tick species. Therefore, the presence of a competent tick vector determines the probability of occurrence of tick-borne piroplasm in an area. To date, *Babesia bovis*, *B. bigemina*, *B. occultans*, *Theileria parva*, *T. annulata*, *T. mutans*, *T. velifera*, *T. taurotragi*, and *T. buffeli* (*T. sergenti*/*T. orientalis*) have been detected in cattle in SSA [6,7]. However, due to the diversity of ecosystems within SSA, tick species distribution varies from one region to another [8] and with animal movements and climate change, tick distribution is expected to change overtime. Understanding and regularly updating the ticks and piroplasms occurring in each part of SSA is important for developing control strategies. 

Burkina Faso is estimated to have the fourth largest cattle population in West Africa. The estimated 9.1 million-heads cattle population raised all over the country accounts for 36–40% of the value-added agriculture, and ranks third in export products [9,10,11]. Tick vectors of piroplasms have been reported all over the country [12,13,14,15,16] and farmers recognized tick infestation as a major constraint to cattle productivity [12,17]. Yet, a limited number of studies has investigated the epidemiology of piroplasmosis in the country. The abundance, annual variation, infestation patterns of ticks, as well as the occurrence of *B. bovis*, *B. bigemina*, *T. annulata*, *T. mutans*, *T. velifera* infections in cattle have been confirmed in southern, western, eastern or central Burkina Faso [13,16,18,19,20]. However, information about occurrence of tick infestation and bovine piroplasm infections in the northern region of the country is not available. This area, called the Sahel region, holds the highest number of cattle. Livestock herding as the main activity contributes up to 69% of the rural population revenue [9,10,11]. The lack of data on the exposure of Sahelian cattle to piroplasmosis hinders efforts to improve animal health and people’s livelihood in that region. With that background, this study was carried out to provide updated data on the situation of tick infestation and bovine piroplasmosis in cattle of the Burkina Faso Sahel region. Cattle herds in two provinces, namely, Oudalan and Séno, were sampled to determine the extent of tick infestations, the occurring tick species, and to detect and characterize tick-borne piroplasms.

## 2. Results

### 2.1. Characteristics of Surveyed Farms and Examined Cattle 

A total of 22 herds and 174 cattle in 6 different villages located in Oudalan and Séno provinces, Burkina Faso, were covered by this study (Figure 1). In Oudalan, 17 farms from 5 different sites and in Séno, 5 farms from one village, were enrolled. All farm owners were male, practiced both crop farming and animal husbandry, and belonged to the Fulani (86.4%), Sonrhai (9.1%), or Touareg (4.6%) ethnic group. Within each farm, 1 to 29 cattle were randomly selected and examined. Detailed distribution of the variables describing the farms is summarized in Table 1.

All examined animals were Zebu (*Bos indicus*) and belonged to the Zebu peuhl breed. Overall, 134 and 40 Zebu of various ages were examined in Oudalan and Séno, respectively. Most cattle belonged to Fulani farmers (71.8%), were female (82.2%), and were apparently healthy (98.8%). Detailed distribution of the variables describing the examined animals is summarized in Table 2.

### 2.2. Tick Species Occurrence and Distribution

Ticks were found on at least one animal in 81.8% (18/22) of investigated farms. The percentage of infested farms varied based on the province. All farms (5/5) in Séno were tick-infested, whereas 76.5% (13/17) were in Oudalan.

The majority of examined cattle (56.3%, 98/174) were tick-free. Tick-infested cattle were mostly found in Séno where 90% (36/40) of animals harbored at least one tick. In Oudalan, 21.9% (40/134) of the cattle were tick-infested. From tick-infested cattle, a total of 144 ticks were collected and later identified as belonging to five species, namely *Hyalomma rufipes*, *H. impeltatum*, *H. impressum*, *Rhipicephalus guilhoni*, and *R. evertsi evertsi* (Figure 2).

*Hyalomma rufipes* was the most abundant tick species (127/144) collected in both provinces and present in all tick-infested farms. *Hyalomma impeltatum* which was the second most abundant species, and *H. impressum* were sampled only in Oudalan. *Rhipicephalus guilhoni* and *R. evertsi evertsi* were found only in Séno. All ticks were at the adult stage and males (109) were more frequent than females (35). The distribution of the tick species across the study areas and cattle farms is presented in Table 3.

### 2.3. Tick-Borne Piroplasm Occurrence and Distribution

Combining species-specific and genera-specific nPCR assays, 23 cattle (23/174; 13.2%) covering more than half of the surveyed farms (12/22; 54.5%) were positive for one or more piroplasms. In Oudalan, 9.7% (13/134) of cattle distributed in 7 different farms (7/17; 41.2%) were positive, whereas in Séno, 25% (10/40) were positive and all farms (5/5) had at least one infected animal. Piroplasm species detected by species-specific primers were *B. bigemina*, *T. annulata*, and *T. mutans*, while genera-specific primers combined with sequence analysis detected *B. occultans* and *T. velifera*. None of the samples were positive for *B. bovis, T. parva*, *T. taurotragi*, or *T. orientalis*. 

*Theileria mutans*, the most frequent pathogen, was detected in 12/174 animals, 4 of which were located in Oudalan and the remaining 8 were from Séno. In both provinces, *T. mutans* was found in 4 different farms. Meanwhile, *B. bigemina* was detected in 5 cattle (5/174) located in 3 farms from Oudalan. Three *T. annulata*-positive samples (3/174) were found in a farm from Oudalan, whereas *B. occultans* was detected in one sample (1/174) from the same province. *Theileria velifera*, however, was amplified in 3 cattle samples (3/174) obtained in 2 farms of Séno. The distribution of piroplasm and tick species identified in this study along with the pathogens reported in cattle from other provinces [16,18,19] are shown in Figure 3.

### 2.4. Risk Factors for Tick Infestation and Piroplasm Infection among Examined Cattle

The results of risk factor analysis are presented in Table 4. Geographic location at the province, as well as village levels and animal age were significant (*p* < 0.05) explanatory variables for the patterns of tick infestation and piroplasm positivity observed among examined cattle. The ethnic group of the farm owner had a significant effect only for tick infestation. Animal gender did not have any significant effect on the probability of being tickinfested or piroplasm-positive. The tick infestation status of a cattle or its presence in a tick-infested farm were not significantly associated with positivity for piroplasm (*p* > 0.05). Tick-infested cattle (17.1%, 13/76) were as frequently piroplasm-positive as the non–infested (10.2%, 10/98). However, there was no piroplasm in the farms where no ticks were detected. Interestingly, *H. rufipes* was detected in all the farms where piroplasms were detected. Animal health status was excluded from the risk factor analysis for tick infestation and piroplasm positivity because most of the cattle (98.8%) were apparently healthy. Meanwhile, occupation, education level of farm owner, transhumance practice, acaricide use, and animal health care provider were excluded from the risk factor analysis due to many missing values.

The odds ratio (ORs) deduced from the Fisher’s exact test showed that cattle in Séno were more likely to be tick-infested (OR: 20.8, CI = 6.8–85.4, *p* < 0.00001) and piroplasm-positive (OR: 3.1, CI = 1.1–8.5, *p* < 0.05). Adult cattle (>5 years) were more exposed to ticks (OR: 3.3, CI = 1.7–6.5, *p* < 0.001) and more at risk for piroplasm infection (OR: 4.5, CI = 1.6–14.8, *p* < 0.01). Cattle owned by Fulani were more likely (3.2 times on average, CI = 1.5–7.4, *p* < 0.01) to be tick-infested than those owned by Sonrhai and Touareg (Table 4).

### 2.5. Analysis of Tick-Borne Piroplasm Sequences 

Genetic characterization of the piroplasms detected in Oudalan and Séno was performed based on sequences obtained with species-specific primers (*B. bigemina RAP-1a*, *T.*
*annulata Tams1*, *T. taurotragi* 18S rRNA and *T. mutans* 18S rRNA) and genera-specific primers (18 S rRNA (long)). The sequence obtained using *T. taurotragi* 18S rRNA primers was identified as *T. annulata* 18S rRNA in the BLASTn search and later confirmed as such through analysis of clone sequences. The piroplasm species identified in the samples shared between 98.4 and 100% sequence identity with their highest BLASTn matches (Table 5).

For *B. bigemina*, one *RAP-1a* (OK323209, 412 bp) and five 18S rRNA sequences (OK314932 and OK314933, 1396 bp; OK314929, OK314930 and OK314931, 1489 bp) sharing 98.9–99.9% identity were recovered from the samples. The closest matches were isolates obtained from cattle in Tanzania, Uganda, Kenya, Egypt, and Turkey for *RAP-1a* sequence and isolates from the Virgin Islands, Mexico, Spain, Turkey, and Switzerland for 18S rRNA sequences. *Babesia occultans* 18S rRNA sequence (OK314934, 1501 bp) was 100% identical to isolates obtained from cattle in South Africa and Turkey and from *H. marginatum* ticks in Tunisia.

*Theileria annulata* sequences included four *Tams1* (OK323210-OK323213, 452 bp), four 18S rRNA obtained with species-specific primers (OK314935-OK314938, 244 bp), and one 18S rRNA sequence obtained with genera-specific primers (OK314939, 1544 bp). The *Tams1* and 18S rRNA (short) sequences showed genetic diversity with pairwise identities ranging from 96.9 to 98.9 % and 99.2 to 99.6%, respectively. Oudalan *T. annulata* parasites were close to those reported in cattle from Mauritania and Egypt for *Tams1* sequences, and those from Egypt, Italy, Pakistan, Turkey, and India for 18S rRNA sequences.

The *T. velifera* identified in Séno was characterized using six 18S rRNA nucleotide sequences which had two different sizes (OK314940, OK314941, OK314943, 1547 bp; OK314942, OK314944, OK314945, 1546 bp) and shared 98.8–99.9% pairwise identities. They were related to isolates obtained from cattle in Uganda and African buffalo (*Syncerus caffer)* in South Africa. 

Sequences from *T. mutans* species-specific 18S rRNA amplicons (OK323969, 196 bp) were conserved across the study areas. However, sequences obtained using the genera-specific primers showed variations. Ten sequences (OK314946-OK314955), three lengths (1454, 1534, and 1537 bp) and pairwise identities between 98.1 and 99.9 % were recorded. Some of the sequences shared identity with *T. mutans* isolated from cattle in Uganda, while others were rather similar (99.93–100%) to a *T. mutans*-related species called *Theileria* sp. B15a (JN572700) and isolated in African buffalo in South Africa. Interestingly, isolates linked to *Theileria* sp. B15a were only found in Séno, whereas those similar to Uganda *T. mutans* were in both provinces. 

### 2.6. Phylogenies of Tick-Borne Piroplasm from Northern Burkina Faso

Phylogenetic analysis based on the 18S rRNA gene sequence was performed to assess the relationship between the *B. bigemina*, *B. occultans*, *T. annulata, T. mutans*, and *T. velifera* in this study and related bovine *Theileria* and *Babesia* species. Two cladograms, one for *Babesia* species and one for *Theileria* species covering 1396 and 1426 bp, respectively, were constructed.

Although all the *B. bigemina* sequences were in the same section of the tree, they were divided in two clusters, labeled Burkina Faso *B. bigemina* Cluster 1 and Burkina Faso *B. bigemina* Cluster 2. The closest sequence to Cluster 1 was isolated from cattle in Brazil, whereas Cluster 2 neighboring cluster included Turkey isolates from cattle and *R. boophilus annulatus* ticks. The *B. occultans* sequence formed a divergent branch within a clade made of *B. occultans* isolates from other areas (South Africa, Tunisia, Turkey) and *Babesia* sp. Kashi 2 isolated from Chinese cattle (Figure 4).

In the *Theileria* species cladogram, the *T. annulata* sequence was located in a clade with previously published sequences obtained in cattle (India, Iran) and donkey (Turkey). It was clearly different from the *Theileria* sp. Yokoyama recently discovered in Sri Lanka. *Theileria velifera* sequences were all in one clade including sequences originating from both cattle and African buffalo. In contrast, there seemed to be more sequence variation within *T. mutans* 18S rRNA gene sequences. They formed two distinct clades. Height sequences grouped with Kenya *T. mutans* Intona strain, along with isolates originating from Ugandan cattle. The remaining two sequences were in *Theileria* sp. (strain MSD)/*Theileria* sp. B15a clade made of isolates from cattle and African buffalo (Figure 5). 

## 3. Discussion 

Although tick infestation was believed to be low in Sahelian areas, this survey in two provinces of the Burkina Faso Sahel region demonstrated the extent of tick infestation and the presence of pathogenic tick-borne piroplasms among cattle from the area for the first time. Ticks belonging to *Hyalomma* and *Rhipicephalus* genera were widely present in the study areas, with cattle infestation having been influenced by geographic area, animal age, and the ethnic group of the farmer. There is recorded evidence that temperature and rainfall patterns have direct effects on the persistence of permanent populations of ticks, provided that hosts are available at adequate densities [21]. The higher amount of annual rainfall in Séno could explain why higher tick infestation were recorded in that area. Likewise, Ouedraogo et al. [16] surveying cattle in eastern Burkina Faso and northern Benin observed that cattle were more infested by ticks as climatic conditions changed from semi-arid (eastern Burkina Faso) to humid (northern Benin). The effect of animal age on tick infestation could be related to older animals having a longer exposure to ectoparasites. Farming practices may also have an impact on infestation as young animals are generally grazed around the housing, while older stocks use distant pastures or are seasonally moved to other areas when pastures are scant. The effect of farmer ethnicity could be a result of an inevitable sampling bias, as Fulani, being the main cattle herder group in the study areas, owned more than 70% of examined animals. It could also just portray the specificity of Fulani herding practices as they are well-known for moving their stocks to humid areas during dry season, and thus increase their animal exposure to ticks. Another explanation could be that non-Fulani are more likely to use acaracide to protect their cattle, while Fulani often use traditional methods, such as manually removing ticks from their animals every day [12,22,23].

Tick control is usually achieved by using acaricides. A survey of tick control practices in central and southern Burkina Faso [12] showed that 73% (44/60) of farmers used conventional acaricides to protect their cattle from tick infestation. In contrast, in this study, only 9.1% (2/22) of farmers reported using acaricides, 50% (11/22) did not use acaricides and 40.9% (9/22) did not answer the question. Unfortunately, due the high non-response rate, it was not possible to assess the influence of acaricide use on tick infestation and piroplasm infection. Further studies investigating farmer knowledge, attitudes, and practices towards ticks and tick-borne diseases and their impact on tick infestation and tick-borne piroplasm infection in the Sahel region will help in devising appropriate control measures.

Our findings are in agreement with previous surveys [14,15] which reported the occurence of *H. rufipes*, *H. impressum*, *H. impeltatum*, and *R. guilhoni* in the study area. However, this is, to the best of our knowledge, the first confirmation of *R. evertsi evertsi* in the Sahel region. The presence of these ticks in northern Burkina Faso is in agreement with the tick distribution in Sahel and Sahara areas of Africa [8]. The high number and ubiquitous distribution of *H. rufipes* is in accordance with Morel’s [14] report and is explained by the fact that it is the most widespread *Hyalomma* species in Africa and commoner in drier areas [8]. The difference in life cycle and seasonal occurrence between tick species also explain the pattern of abundance of each species. The presence of some species in one but not in the other province could be either related to the impact of microclimate differences on species distribution/abundance or an effect of study design (study period, number of animals examined). The latter seemed more plausible because *R. guilhoni*, which was not found on cattle in Oudalan, could be collected on small ruminants examined during the same period (manuscript in preparation). 

The collected ticks have either a two-host (*H. rufipes*, *R. evertsi evertsi*) or three-host life cycle (*R. guilhoni*, *H. impeltatum*, *H. impressum*) and the preferred hosts of their adult stages are cattle. Except for *R. evertsi evertsi,* the immatures stages (larva, nymph) do not infest cattle and exclusively feed on hares, rodents, or ground-frequenting birds [8,24]. It was therefore unlikely to find larva and nymphs on examined cattle.

The observed effect of location and age on piroplasm positivity is explained by how these variables modulate exposure to tick vectors. The detection of *T. mutans*, *T. velifera, B. bigemina*, and *T. annulata* is consistent with reports that showed the circulation of these pathogens in other provinces in Burkina Faso [16,18,19]. Compared to previous studies, the overall piroplasm infection rates recorded in cattle from Séno and Oudalan were low. This could be explained by the difference in climate between study areas. Previous surveys were performed in the North-Sudanian and South-Sudanian climate zones (annual rain fall > 600 mm) which harbor a more abundant and diversified tick fauna [13,16,20] suggesting that these cattle had a higher tick exposure than those in the Sahel region.

The wide distribution of *T. mutans* agrees with Ouedraogo et al.’s [16] survey which reported that it was the most frequent pathogen. The phylogenetic analysis based on 18S rRNA gene sequences indicated the presence of *T. mutans* Intona strain-like and *Theileria* sp. B15a-like isolates. *Theileria* sp. B15a was first reported in South Africa and classified as *Theileria* sp. (strain MSD) [25]. In the absence of further characterization, it is difficult to know whether *Theileria* sp. B15a /*Theileria* sp. (strain MSD) represents a different species or a *T. mutans* variant found both in cattle and African buffalo. Its presence in the study area calls for further investigation of the diversity of *T. mutans* in livestock and wildlife in Burkina Faso.

Ouedraogo et al. [16] reported *T. velifera* as the second most frequent tick-borne pathogens among cattle in south-eastern Burkina Faso. The frequency of *T. velifera* and *T. mutans*, both being transmitted by *Amblyomma* spp. [6], is generally similar, but this was not the case in our survey. The absence of positive animals in Oudalan and the low infection rate could be partly due to the diagnostic approach used. A species-specific PCR assay may have allowed us to detect more positive cases. Additional molecular surveys are needed to clarify the extent of the prevalence of this protozoon in northern Burkina Faso.

*Babesia bigemina* was invariably reported in all surveys performed in Burkina Faso (Figure 3). Four *B. bigemina* competent vectors, namely *R. boophilus microplus*, *R. b. annulatus*, *R. b. geigyi*, and *R. b. decoloratus*, have been reported in Burkina Faso [13,15,16]. Due to the diversity of ecology of these ticks, competent vectors are found all over the country allowing such distribution of the protozoon. Notably, although previously reported in the region [15], *R. b. decoloratus* (for *B. bigemina*) and *Amblyomma variegatum* (for *T. velifera* and *T. mutans*) were not encountered during our study. This points to the need for a more comprehensive study to update the distribution of competent vector for piroplasms in the Sahel region of Burkina Faso. 

Until recently, the known distribution of *T. annulata* in Africa was limited to eight African countries, namely Mauritania, Morocco, Algeria, Tunisia, Egypt, Sudan, South Sudan, and Ethiopia [26]. This year, two reports added Benin, Burkina Faso [16], and Nigeria [27] to the list. The detection of this parasite in Oudalan is in accordance with the presence of *H. impeltatum*, a well identified competent vector [8]. Our findings bring to four the number of provinces in Burkina Faso where *T. annulata* is reported and indicate the need for investigating it in other provinces.

*Babesia occultans* has not been reported before in cattle in Burkina Faso or in West Africa. It was described for the first time 40 years ago in South Africa [28], with *H. rufipes* as the competent vector. Thereafter, kinetes identified as *B. occultans* were found in *H. rufipes*, *H. truncatum*, *H. impressum*, *H. marginatum*, and *H. impeltatum* in Nigeria [29]. In Tunisia, northern Africa, it was detected by PCR in *H. marginatum* [30]. Other reports located it in Europe [31,32,33,34] or Asia [35,36]. The detection of *B. occultans* in Oudalan expands its distribution and can be linked to the cattle infested with *H. rufipes*, *H. impressum* and *H. impeltatum*.

*Babesia bigemina*, *T. annulata*, *T. mutans*, and *T. velifera* can cause acute or chronic asymptomatic infection [6,7]. The absence of clinical signs in most of the infected cattle may indicate cases of chronic infections. In contrast, *B. occultans* has never been associated with clinical signs in African cattle. Yet, it has caused clinical outbreak of bovine piroplasmosis in Italy [31]. Therefore, similar to the other piroplasms, the occurrence of *B. occultans* in Burkina Faso should not be neglected.

This study had some limitations. The cross-sectional design, the number/distribution of study sites, the convenience sampling of farms, and the small number of samples may have prevented us from grasping the full extent of tick and piroplasm species distributions in the Sahel region. Nevertheless, through the combination of interviews, species-specific and genera-specific assays, and systematic inspection of animals for ticks, this survey provided novel information that will help animal health stakeholders in the area.

## 4. Materials and Methods

### 4.1. Ethical Statement

All cattle owners were informed about and approved the objectives of the study. Animal restrain, blood sampling, and tick collection were performed with the help of cattle keepers and with care to minimize pain and discomfort. All operations were carried out in accordance with the ethical guidelines for the use of animal samples established by Obihiro University of Agriculture and Veterinary Medicine (animal experiment approval number: 19–74; DNA experiment approval number: 1706, 1705-4, 1704-4).

### 4.2. Study Area

The Sahel region of Burkina Faso consists of four provinces and covers 35,360 km^2^. The region is overwhelmingly rural with most of the population involved in animal husbandry and crops farming. Two different climates are observed: the hot semi-arid climate characterized by annual average temperature and rainfall of 29.0 °C and 473 mm and the desert climate which has virtually no rainfall with an average temperature of 29.1 °C and annual average precipitation of 382 mm [37]. Two seasons are present: the dry season which lasts 6–8 months and the rainy season which occurs from May/June to September/October with rainfall peaking in August. The vegetation is characterized by shrubby steppes, scattered tall trees, and grassland. Although pastoral, semi-intensive, and intensive cattle production systems are observed in the region, 72% of cattle are kept under the traditional agropastoral system [10]. Cattle are fed on natural pastures around the homestead and bred for meat and milk production. In this study, agropastoralist cattle herds were investigated in two provinces of the region: Oudalan province (Oudalan) which has a desert climate, and Séno province (Séno) which has a semi-arid climate.

### 4.3. Study Design

The field survey was conducted from 14 to 20 October 2016 and included farmer interviews and collection of ticks and cattle blood. Sampling sites and herders enrolled in the study were selected among cattle farmers involved in VSF livestock development programs. Prior to field activities, sensitization meetings were conducted by VSF local partners. At each study site, several farms were recruited. In each farm, herders were interviewed and several cattle were selected for examination. Herd owners were interviewed on their profile (name, age, ethnic group, and occupation) and the characteristics of the farms (animal species, breeds, herd size, feeding and animal health care practices). The questionnaire was administered individually in the local language and according to the convenience of the respondent. Afterwards, 10 percent of the cattle herd including representative of different age groups was randomly selected for examination and sample collection. The field survey team included two veterinarians, one laboratory technician, and one representative of VSF local partners.

### 4.4. Tick and Blood Samples

Ticks and blood samples were collected from male and female cattle of various ages. In detail, animal age was estimated based on information provided by the owner. Then, each animal was checked for tick infestation and health status (apparently healthy or sick). Animal health status was decided based on general appearance, clinical examination by a veterinarian, farmer’s information, and when applicable, clinical signs were recorded. When present, ticks were picked with forceps and stored in a labeled glass vial containing 70% ethanol. Ticks collected in the same farm were stored in one vial. Approximately 5 mL of whole blood was aseptically collected from the jugular vein of each examined animal using EDTA-coated vacutainer tubes. During blood collection, disposable gloves, sterile needles, tubes, vacuum blood collection system with holder, and 70% isopropyl alcohol-soaked cotton pads were used to avoid contamination. Blood samples were kept on ice and tick samples were kept at room temperature until transported to the Laboratoire National d’Elevage (LNE), Ouagadougou, Burkina Faso. At the LNE, ticks and blood samples were stored at 4 °C pending species identification and DNA extraction, respectively. 

Pathogen genomic DNA was extracted from each sample using 200 μL of blood sample and a commercial kit (QIAamp DNA Blood Mini-Kit, Hilden, Germany). Blood DNA samples and tick vials were then transported from the LNE to the National Research Center for Protozoan Diseases, Obihiro University of Agriculture and Veterinary Medicine, Obihiro, Japan (NRCPD). DNA samples were stored at −30 °C until molecular screening. For tick samples, species identification and morphological classification were carried out using a binocular microscope (Olympus SZX16, Japan) and previously established standard taxonomic keys [8]. The tick samples were stored at 4 °C.

### 4.5. Molecular Detection of Tick-Borne Piroplasms

The detection of *Babesia* and *Theileria* species was done in two steps. First, all cattle blood DNA samples were screened for *B. bovis*, *B. bigemina*, *T. parva*, *T. annulata*, *T. orientalis*, *T. taurotragi*, and *T. mutans*, using previously described species-specific nested PCR (nPCR) assays. Primers amplifying the partial sequences of *B. bovis spherical body protein-2* (*SBP-2*) [38], *B. bigemina rhoptry-associated protein-1a* (*RAP-1a*) [39], *T. parva* 18S rRNA, *T. taurotragi* 18S rRNA, *T. mutans* 18S rRNA [40], *T. orientalis major piroplasm surface protein* (*MPSP*) [41], and *T. annulata major merozoite surface antigen 1* (*Tams1*) [42,43,44] genes were employed. 

After detection of piroplasm known to cause diseases, to confirm the results and identify other species, all samples were screened with a genera-specific nPCR assay. PCR primers amplifying a 1500–1600 bp fragment of *Babesia* and *Theileria* species 18S rRNA [45] genes were used. 

PCR assays were setup as follows, initial amplification was done in a total volume of 10 μL including 1.5 μL of DNA template, 0.2 μL (10 μM) of each primer, 1 μL of 10× ThermoPol reaction buffer, 0.2 μL of dNTP solution mix (100 μM each), 0.05 μL of Taq DNA Polymerase (New England Biolabs, MA, USA), and 6.85 μL of double-distilled water. The second PCR was done using 1.5 μL of DNA template obtained from the first PCR amplification. Except for denaturation and extension temperatures and times which were changed to fit the PCR protocol for Taq DNA Polymerase, thermocycling conditions were as in referenced publications. The DNA of *B. bigemina* (Argentina strain), *B. bovis* (Texas strain), *T. parva* (Muguga G6, ILRI), *T. annulata* (Ankara C9, Edinburgh University), and cattle DNA sample positive for *T. orientalis*, *T. taurotragi*, and *T. mutans* [46] were used as positive controls in the corresponding PCR assays. Double-distilled water was the negative control. Final amplification products were electrophoresed on 1.5% agarose gel, stained in ethidium bromide solution (Nacalai Tesque, Japan), then visualized under UV transilluminator (Printgraph AE-6905CF, Atto, Japan). Positive samples were identified by a band of similar size with the positive control.

### 4.6. Sequencing of Piroplasm DNA

All positive samples were used as templates for genetic characterization of detected piroplasm. In detail, positive samples were subjected to a new set of nPCRs to obtain larger volume of amplicons. The first PCR was conducted in a 10 μL-reaction mixture composed of 1 μL of DNA template, 1 μL (10 μM) of each primer, 1 μL of 10× Ex Taq buffer, 1 μL of dNTP (200 μM each), 0.1 μL of Ex Taq polymerase (Takara, Japan), and 4.9 μL of double-distilled water. The nested PCR was done with 3 μL of DNA template of the first PCR amplification and the reagents mentioned above adjusted for a 50 μL-reaction. Thermocycling conditions were as in referenced manuscripts. Amplicons from the second PCR were purified from gel using QIAquick Gel Extraction Kit (QIAGEN GmbH, Germany) and sequenced with the amplification primers. Resulting DNA sequences were analyzed, checked for heterogeneous base-calling using Mixed Sequence Reader web-based program [47], and compared to each other. When PCR products from different samples showed the same sequence, only one specimen was considered for further analysis. 

Except for the products of *T. mutans* 18S rRNA nPCR assay, amplicon representatives of the different types of sequences identified in the study samples were cloned in pGEM-T Easy Vector (Promega, USA) and sequenced as previously reported [48]. Clone sequences were manually trimmed to remove the vector sections and compared to direct sequences results. 

All the sequences were obtained by performing Sanger sequencing using the BigDye^™^ Terminator v3.1 Cycle Sequencing Kit (Applied Biosystems, USA) and ABI Prism 3100 Genetic Analyzer (Applied Biosystems) at the NRCPD, Obihiro, Japan.

All the sequences obtained in this study were registered in the NCBI GenBank database. The accession numbers are: OK323209 (*B. bigemina*
*RAP-1a*); OK314929-OK314933 (*B. bigemina* 18S rRNA); OK314934 (*B. occultans* 18S rRNA); OK323210-OK323213 (*T. annulata Tams1);* OK314935-OK314939 (*T. annulata* 18S rRNA); OK314940-OK314945 (*T. velifera* 18S rRNA); OK323969, OK314946-OK314955 (*T. mutans* 18S rRNA).

### 4.7. Analysis of DNA Sequences

Microorganism DNA sequences were first analyzed using the nucleotide BLAST algorithm of NCBI GenBank database (BLASTn) to confirm their identity. Pathogen species identity was based on the closest BLASTn match to sequences available in the GenBank. When several sequences were obtained for the same target, the EMBOSS NEEDLE software (http://emboss.bioinformatics.nl/cgi-bin/emboss/needle (accessed on 20 October 2021)) was used to assess the percent identities between the sequences. The 18S rRNA sequences (1500–1600 bp) of all piroplasm identified in the survey were aligned with references species and a phylogenetic tree was constructed. Multiple sequence alignments were performed and tested with the web-based program GUIDANCE 2 [49] and phylogenetic trees were inferred by the maximum likelihood method using MEGA version X software [50].

### 4.8. Statistical Analysis

Descriptive statistics were computed to assess the features of the farms and animals enrolled in the study. Tick infestation and pathogen infection rates were calculated. Risk factors for tick infestation and pathogen positivity were identified by performing the Fisher’s exact test for Count Data of tidyverse package [51] in R statistical software (https://www.R-project.org/ (accessed on 20 October 2021)). A *p* value < 0.05 was considered statistically significant.

## 5. Conclusions

This study confirmed the presence of five tick species and assessed the extent of tick infestation in cattle from two provinces in northern Burkina Faso. Five piroplasm species were detected in cattle: *B. bigemina*, *T. annulata*, *T. mutans*, and *T. velifera*, which had not been previously reported in the region and *B. occultans*, which had not been detected in cattle in West Africa before. These findings open new perspectives on TTBDs in the Sahel region of Burkina Faso and suggest that veterinarians should be aware of piroplasmosis when treating cattle in the area. 

## Figures and Tables

**Figure 1 pathogens-11-00031-f001:**
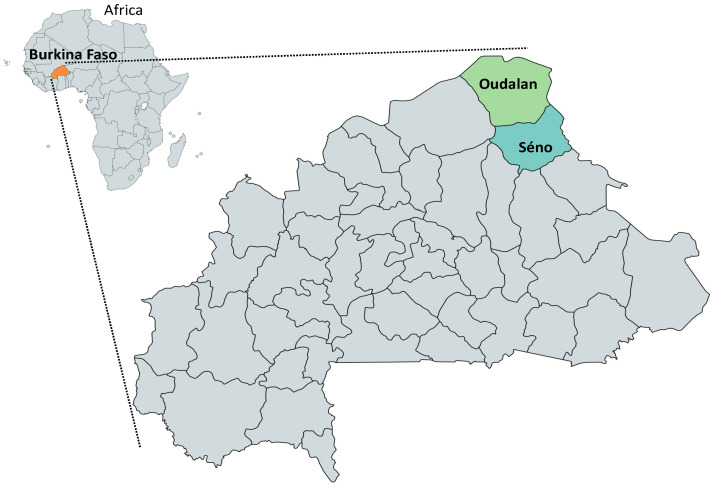
Map of Burkina Faso showing the study provinces. Cattle farms located in Oudalan and Séno provinces in the Sahel region of Burkina Faso were surveyed.

**Figure 2 pathogens-11-00031-f002:**
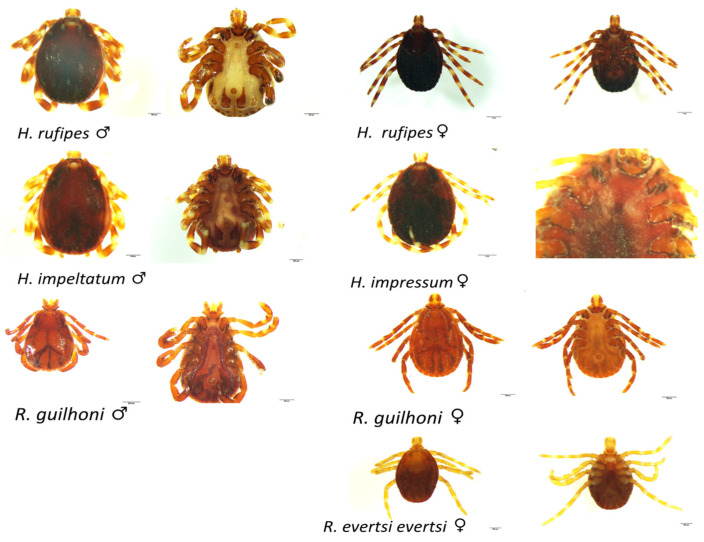
Dorsal and ventral views of the tick specimen collected on cattle in Oudalan and Séno provinces, northern Burkina Faso. *H*.: *Hyalomma*; *R*.: *Rhipicephalus;* ♂: adult male, ♀: adult female. Black bar: 1 mm (*H. rufipes* ♀; *H. impressum* dorsal view), 200 μm (*H. rufipes* ♂ ventral view; *H. impressum* ventral view; *R. guilhoni* ♂), 500 μm (*H. rufipes* ♂ dorsal view; *H. impeltatum*; *R. evertsi evertsi*, *R. guilhoni* ♀).

**Figure 3 pathogens-11-00031-f003:**
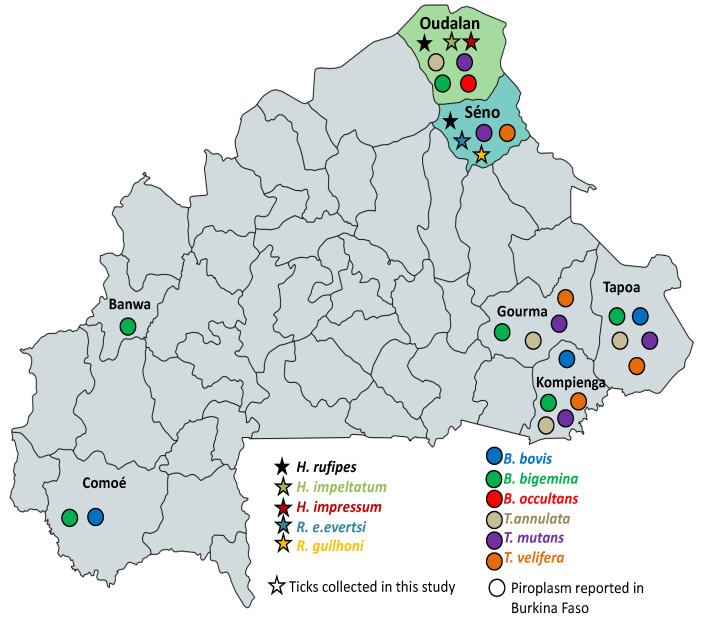
Map of Burkina Faso with the distribution of bovine tick-borne piroplasm species reported in cattle in different provinces. Tick species and pathogens identified in Oudalan and Séno provinces (this study) and piroplasm infections reported in other provinces [16,18,19] are presented. The stars indicate ticks and circles represent piroplasms. Colors indicate pathogen or tick species. B.: *Babesia*; T.: *Theileria*; H.: *Hyalomma*; R.: *Rhipicephalus*; e.: *evertsi*.

**Figure 4 pathogens-11-00031-f004:**
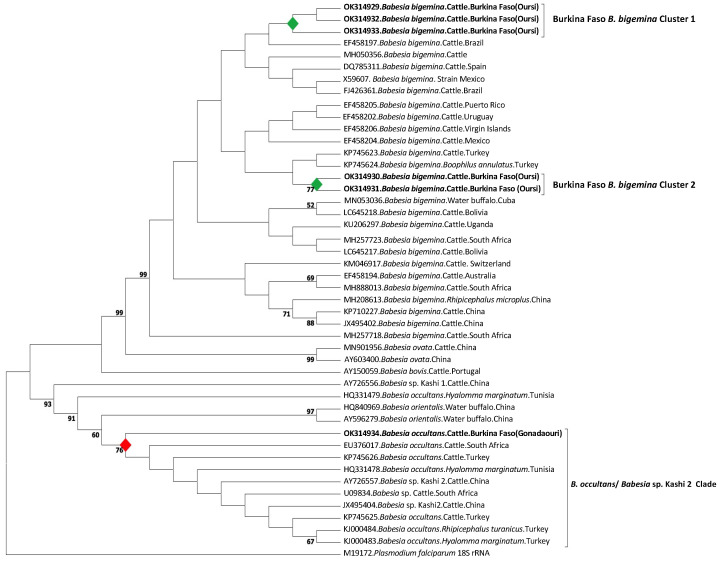
Phylogenetic tree showing the relationship between the *B. bigemina* and *B. occultans* 18S rRNA sequence variants identified in this study with other bovine and buffalo *Babesia* species as estimated by maximum likelihood analysis. This cladogram covers 1396 bp and 46 nucleotide sequences and was constructed in MEGA X using Tamura-Nei model and a discrete Gamma distribution to model evolutionary rate differences among sites (+G). *Plasmodium falciparum* 18S rRNA sequence was used as outgroup. Bootstrap values are shown as percentages at nodes based on 1000 replicates, and values lower than 50% were omitted. The sequences obtained from the current study are shown in bold.

**Figure 5 pathogens-11-00031-f005:**
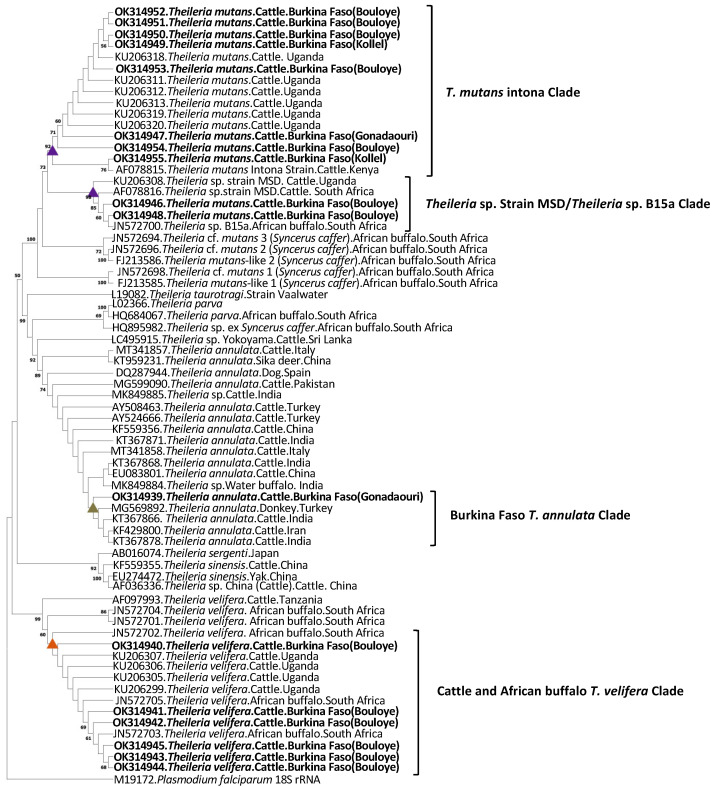
Cladogram showing the relationship between the *T. annulata*, *T. mutans*, and *T. velifera* 18S rRNA sequence variants identified in this study with other bovine and buffalo *Theileria* species as estimated by maximum likelihood analysis. The phylogenetic analysis covers 1426 bp and 69 nucleotide sequences and was constructed in MEGA X using the Tamura 3 parameter model and a discrete Gamma distribution to model evolutionary rate differences among sites (+G). The rate variation model allowed for some sites to be evolutionarily invariable (+*I*). *Plasmodium falciparum* 18S rRNA sequence was used as outgroup. Bootstrap values are shown as percentages at nodes based on 1000 replicates, and values lower than 50% were omitted. The sequences obtained from the current study are shown in bold.

**Table 1 pathogens-11-00031-t001:** Characteristics of the cattle farms surveyed in Oudalan and Séno provinces, northern Burkina Faso.

Variables	Category	Frequency per Province (%)		
		Oudalan (N = 17)	Séno (N = 5)	Total (N = 22)
Location	In Oudalan			
	Kollel	7 (41.2)	-	7 (31.8)
	Gonadaouri	2 (11.8)	-	2 (9.1)
	Gorom gorom	5 (29.4)	-	5 (22.7)
	Tin ediar	1 (5.9)	-	1 (4.6)
	Oursi	2 (11.8)	-	2 (9.1)
	In Séno			
	Bouloye	-	5 (100)	5 (22.7)
Ethnic group of owner	Fulani	14 (82.4)	5 (100)	19 (86.4)
	Sonrhai	2 (11.8)	0	2 (9.1)
	Touareg	1 (5.9)	0	1 (4.6)
Main occupation of owner	Crop farming	9 (52.9)	5 (100)	14 (63.6)
	Livestock farming	3 (17.6)	0	3 (13.6)
	Night watchman	1 (5.9)	0	1 (4.5)
	No answer	4 (23.5)	0	4 (18.2)
Secondary occupation of owner	Crop farming	3 (17.6)	0	3 (13.6)
	Livestock farming	9 (52.9)	5 (100)	12 (63.6)
	Crop and livestock	1 (5.9)	0	1 (4.6)
	No answer	4 (23.5)	0	4 (18.2)
Education level of owner	No formal education	5 (29.4)	0	5 (22.7)
	Fulfulde courses	1 (5.9)	0	1 (4.6)
	Koranic school	3 (17.6)	0	3 (13.6)
	Primary school	3 (17.6)	1 (20)	4 (18.2)
	Secondary school or higher	1 (5.9)	0	1 (4.6)
	No answer	4 (23.5)	4 (80)	8 (36.4)
Transhumance practice	No	6 (35.3)	0	6 (27.3)
	Yes	6 (35.3)	1 (20)	7 (31.8)
	No answer	5 (29.4)	4 (80)	9 (40.9)
Acaricide usage	Yes	1 (5.9)	1 (20)	2 (9.1)
	No	11 (64.7)	0	11 (50)
	No answer	5 (29.4)	4 (80)	9 (40.9)
Animal health care provider	CBAHW or veterinarian	12 (70.6)	1 (20)	13 (59.1)
	No answer	5 (29.4)	4 (80)	9 (40.9)
Cattle breeds	Zebu only	16 (94.1)	5 (100)	21 (95.5)
	Zebu, taurine and crossbred	1 (5.9)	0	1 (4.6)
No. of cattle examined	1–5	7 (41.2)	2 (40)	9 (40.9)
	6–10	8 (47.1)	3 (60)	11 (50)
	11–15	1 (5.9)	0	1 (4.6)
	>16	1 (5.9)	0	1 (4.6)

N = Number of cattle farms surveyed; GPS coordinates of the study sites: Kollel (14°33′08″ N 0°25′33″ W), Gonadaouri (14°44′51″ N 0°23′31″ W), Gorom gorom (14°26′48″ N 0°13′50″ W), Tin ediar (14°41′26.1″ N 0°36′41″ W), Oursi (14°40′31″ N, 0°27′41″ W), Bouloye (14°03′00″ N, 0°03′00″ W); CBAHW = community based animal health worker.

**Table 2 pathogens-11-00031-t002:** Characteristics of the cattle (*n* = 174) examined in Oudalan and Séno provinces, northern Burkina Faso.

Variables	Category	Frequency per Study Province (%)		
		Oudalan (*n* = 134)	Séno (*n* = 40)	Total (*n* = 174)
Location	In Oudalan			
	Kollel	27 (20.1)	-	27 (15.5)
	Gonadaouri	19 (14.2)	-	19 (10.9)
	Gorom gorom	39 (29.1)	-	39 (22.4)
	Tin ediar	29 (21.6)	-	29 (16.7)
	Oursi	20 (14.9)	-	20 (11.5)
	In Séno			
	Bouloye	-	40 (100)	40 (22.9)
Ethnic group of owner	Fulani	85 (63.4)	40 (100)	125 (71.8)
	Sonrhai	20 (14.9)	0	20 (11.5)
	Touareg	29 (21.6)	0	29 (16.7)
Age group	Suckling	29 (21.6)	0	29 (16.7)
	Weaned immature	57 (42.5)	13 (32.5)	70 (40.2)
	Adult	48 (35.8)	27 (67.5)	75 (43.1)
Gender	Male	27 (20.2)	4 (10)	31 (17.8)
	Female	107 (79.8)	36 (90)	143 (82.2)
Health status	Not sick	132 (98.5)	40 (100)	172 (98.8)
	Sick	2 (1.5)	0	2 (1.2)

*n* = Number of cattle; suckling (<1 year); weaned immature (1 < X < 4 years); adult (>4 year).

**Table 3 pathogens-11-00031-t003:** Species, gender, and distribution of ticks collected from cattle in Oudalan and Séno provinces, northern Burkina Faso.

Tick Species	No. of Specimens Collected (Male/Female)			No. of Infested Farms		
	Oudalan (*n* = 134)	Séno (*n* = 40)	Total (*n* = 174)	Oudalan (N = 17)	Séno (N = 5)	Total (N = 22)
*H.rufipes*	84 (70/14)	43 (28/15)	127 ( 98/29)	12	5	17
*H.impressum*	2 (0/2)	-	2 (0/2)	2	-	2
*H.impeltatum*	9 (9/0)	-	9 (9/0)	1	-	1
*R. evertsi evertsi*	-	1 (0/1)	1(0/1)	-	1	1
*R. guilhoni*	-	5 (2/3)	5(2/3)	-	3	3
Total	95 (79/16)	49 (30/19)	144 (109/35)	13	5	18

Specimen collected (male/female): For each tick species, number of specimens that was collected from examined cattle; infested farm: farm in which the tick species was encountered on at least one cattle; *n*: number of cattle that were examined in each province, N: number of farms in which cattle were examined; *H*.: *Hyalomma*; *R*.: *Rhipicephalus.*

**Table 4 pathogens-11-00031-t004:** Explanatory variables for tick infestation and piroplasm infection among examined cattle.

Parameter	Variable	OR	95% CI	*p* Value
Tick infestation	**Province**	20.8	6.8–85.4	**7.19 × 10^−12^**
	**Ethnic group of owner**	3.2	1.5–7.4	**2.04 × 10^−3^**
	**Age**	3.3	1.7–6.5	**2.03× 10^−4^**
	Gender	0.5	0.2–1.1	7.59 × 10^−2^
Piroplasm positivity	**Province**	3.1	1.1–8.5	**1.75 × 10^−2^**
	Ethnic group of owner	1.5	0.5–5.4	6.20 × 10^−1^
	**Age**	4.5	1.6–14.8	**2.61 × 10^−3^**
	Gender	1.0	0.2–3.3	1
	Tick infestation	1.8	0.7–4.9	2.59 × 10^−1^

OR: odds ratio, CI: confidence interval, lower and upper values; statistically significant variables are in bold.

**Table 5 pathogens-11-00031-t005:** BLASTn results for Piroplasmida sequences obtained from cattle blood samples in Oudalan and Séno provinces, northern Burkina Faso.

DNA Sequences					Highest BLASTn Match	
Pathogen	Target Gene	Accession No.	Length (bp)	Study Sites	GenBank ID (Origin)	% Identity
*B. bigemina*	*Rap-1a*	OK323209	412	Gorom gorom (Oudalan)	MG210824, MN807310, MN807308, MN807306 (Tanzania); MG426198, MG426199, MG426201, MG426202 (Uganda); KP347559 (Kenya); KF192811(Egypt); KT220512 (Turkey)	99.76
	18 S rRNA (long)	OK314932	1396	Oursi (Oudalan)	EF458206 (Virgin Islands); DQ785311 (Spain); X59607, X59604 (strain Mexico)	99.93
	18 S rRNA (long) 18 S rRNA (long)	OK314933	1396 1489	Oursi (Oudalan) Gonadaouri (Oudalan)	EF458206 (Virgin Islands); DQ785311 (Spain); X59607, X59604 (strain Mexico) DQ785311 (Spain); X59607, X59604 (strain Mexico)	99.86
		OK314929				99.6
		OK314930	1489 1501		KP745623 (Turkey); KM046917 (Switzerland); DQ785311 (Spain); X59607, X59604 (strain Mexico)	99.33
		OK314931			KP745623 (Turkey); KM046917 (Switzerland); DQ785311 (Spain); X59607, X59604 (strain Mexico) EU376017 (South Africa); HQ331478 (Tunisia); KP745626 (Turkey)	98.99
		OK314934				100
*B. occultans*	*Tams 1a*	OK323210	452	Gonadaouri (Oudalan)	AF214854, AF214824, AF214823 (Mauritania)	100
*T. annulata*	*Tams 1a* 18 S rRNA (short)	OK323212	452 244	Gonadaouri (Oudalan) Gonadaouri (Oudalan)	AF214854, AF214824, AF214823 (Mauritania) AB917279 (Egypt)	98.67
		OK323213				98.44
		OK323211				99.78
		OK314935			MG599090 (Pakistan)	99.59
	18 S rRNA (short) 18 S rRNA (long)	OK314936	244 1544	Gonadaouri (Oudalan) Gonadaouri (Oudalan)	MT341858, MT341857 (Italy); MT318160 (Pakistan); MN227666 (Egypt), MK849884 (*Theileria* sp. India), KT367871, KT367868 (India)	99.59
	18 S rRNA (short) 18 S rRNA (long) 18 S rRNA (long)	OK314937	244 1544 1547	Gonadaouri (Oudalan) Gonadaouri (Oudalan) Bouloye (Séno)	MT341858, MT341857 (Italy); MT318160 (Pakistan); MN227666 (Egypt), MK849884 (*Theileria* sp. India), KT367871, KT367868 (India) MT341858 (Italy); AY524666, AY508463, MG569892 (Turkey); MK849884 (*Theileria* sp. India); KT367871, KT367868, KT367866 (India)	99.17
		OK314938			MT341858, MT341857 (Italy); MT318160 (Pakistan); MN227666 (Egypt), MK849884 (*Theileria* sp. India), KT367871, KT367868 (India) MT341858 (Italy); AY524666, AY508463, MG569892 (Turkey); MK849884 (*Theileria* sp. India); KT367871, KT367868, KT367866 (India) KU206298, KU206299, KU206300, KU206301, KU206302, KU206303, KU206304, KU206305, KU206306, KU206307 (Uganda); JN 572705 (South Africa)	99.17
		OK314939				100
		OK314940				100
	18 S rRNA (long) 18 S rRNA (short)	OK314941	1547 1546	Bouloye (Séno) Tin eddiar Gorom gorom (Oudalan) Bouloye (Séno)	KU206298, KU206299, KU206300, KU206301, KU206302, KU206303, KU206304, KU206305, KU206306, KU206307 (Uganda); JN 572705 (South Africa) MT250263 (Burkina Faso); MN726650, MN726649 (Tanzania); MK481006 (South Africa), MN124094 (Cameroon); MH424330 (Guinea); KU206309, KU206310 (Uganda); FJ869899, FJ869898 (Mozambique)	99.87
*T. velifera*	18 S rRNA (long) 18 S rRNA (short) 18 S rRNA (long)	OK314943	1547 1546 1546 196	Bouloye (Séno) Tin eddiar Gorom gorom (Oudalan) Bouloye (Séno) Bouloye (Séno)	KU206298, KU206299, KU206300, KU206301, KU206302, KU206303, KU206304, KU206305, KU206306, KU206307 (Uganda); JN 572705 (South Africa) MT250263 (Burkina Faso); MN726650, MN726649 (Tanzania); MK481006 (South Africa), MN124094 (Cameroon); MH424330 (Guinea); KU206309, KU206310 (Uganda); FJ869899, FJ869898 (Mozambique) JN572700 (*Theileria* sp.B15a; South Africa)	99.61
		OK314942				99.68
		OK314944				99.42
		OK314945	1546 196 1534			99.42
		OK323969				100
		OK314946				100
*T. mutans*	18 S rRNA (long)	OK314948	1534 1537	Bouloye (Séno) Gonadaouri (Oudalan)	JN572700 (*Theileria* sp.B15a; South Africa) KU206311, KU206312, KU206313, KU206314, KU206315, KU206316, KU206319, KU206320 (Uganda)	99.93
	18 S rRNA (long)	OK314950	1534 1537 1537 1537	Bouloye (Séno) Gonadaouri (Oudalan) Kollel (Oudalan)	JN572700 (*Theileria* sp.B15a; South Africa) KU206311, KU206312, KU206313, KU206314, KU206315, KU206316, KU206319, KU206320 (Uganda) KU206311, KU206312, KU206313, KU206314, KU206315, KU206316, KU206319, KU206320 (Uganda)	99.8
	18 S rRNA (long)	OK314951	1534 1537 1537 1537 1537 1537 1537	Bouloye (Séno) Gonadaouri (Oudalan) Kollel (Oudalan) Kollel (Oudalan)	JN572700 (*Theileria* sp.B15a; South Africa) KU206311, KU206312, KU206313, KU206314, KU206315, KU206316, KU206319, KU206320 (Uganda) KU206311, KU206312, KU206313, KU206314, KU206315, KU206316, KU206319, KU206320 (Uganda) KU206311, KU206312, KU206313, KU206314, KU206315, KU206316, KU206319, KU206320 (Uganda)	99.74
		OK314952				99.54
		OK314953	1537 1537 1537 1454		KU206311, KU206312, KU206313, KU206314, KU206315, KU206316, KU206319, KU206320 (Uganda)	99.48
		OK314954				99.41
		OK314947				100
		OK314949				99.8
		OK314955				99.38
				Kollel (Oudalan)		
				Kollel (Oudalan)		

B.: *Babesia*, T.: *Theileria*.

## Data Availability

The datasets generated during and/or analyzed during the current study are available from the corresponding author on reasonable request.

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
