# Peer review of "A Survey of Tick Infestation and Tick-Borne Piroplasm Infection of Cattle in Oudalan and Séno Provinces, Northern Burkina Faso"

_pathogens, 2021, doi:10.3390/pathogens11010031_

Round 1
Reviewer 1 Report
- Abstract: The length of the abstract should be reduced to a more concise graph.
- Methods: There should have a correct description regarding the animal and DNA experiments.
Author Response
Dear Referee #1, we are grateful for your comments and suggestions. Following your advice, the manuscript has been through editing of English language and style. Other referees’ suggestions concerning English language and style were also included in the revised manuscript.
Response to Comments and Suggestions for Authors
1. Abstract: The length of the abstract should be reduced to a more concise graph.
Response: In line with your suggestion, the abstract has been shortened from 378 words to 335 words. Two sentences were removed and the content was rearranged to be concise and precise.
2. Methods: There should have a correct description regarding the animal and DNA experiments.
Response: Additional information was added to the Methods section. Animal sampling description and DNA sequencing experiments were presented with more details.
Reviewer 2 Report
The manuscript can be considered for publication in the pathogens journal after minor revision.

Reviewer 3 Report
The manuscript reports a survey study to establish the occurrence and prevalence of piroplasms infections in cattle. The manuscript can be classified in big part into a number of publications duplicating a similar scheme (there is a pathogen, a host, we use PCR, BLAST, we obtain sequences, we make a phylogenetic tree). In general article is well written. The data are certainly valuable, however in local interest mainly. Title adequately reflects the subject of the study. The research design and execution is of good standard. The analysed problems are clearly presented. All methods are correctly used.
Some specific comments (see below):
Abstract: - The abstract does not mention an interesting factor raised in the article - the influence of the ethnic group on tick infestation, there is lack of information on the season.
49 50 – “each piroplasm is associated with particular tick species” – it is not completely true, a supposed to write “many piroplasms are associated”
76 – 80 – I suppose to move that information to acknowledgements.
Table 2 - divide the locations between Oudalan and Seno in the "terms" column. These names are unknown to people not familiar to the details of Burkina Faso's geography.
Table 5 – add hosts (if possible) in the column Highest Blastn match.
2.5. Analysis of (…) sequences - a possible objection is the lack of the obtained sequences in the "results" section, or the accession number under which they were placed in GenBank.Thus, it is impossible for the reader to compare them with other sequences on his own.
283 – the influence of an ethnic group on the level of infection in farmed animals is an extremely interesting topic, rarely discussed in publications. Here is just one of the many factors discussed, but it is worth adding a few details directly affecting the number of ticks.
Detection of tick-borne piroplasms - why were traditional blood smears, stained with Giemsa, not made? It is an inexpensive and quick diagnostic method. In addition to the detection of the infection, it also allows you to estimate the intensity.
Reviewer 4 Report
I reviewed the article titled: “A Survey of Tick Infestation and Tick-borne Piroplasm Infection of Cattle in Oudalan and Séno Provinces, Northern Burkina Faso” and I found it incomplete. The authors have investigated the presence of tick-borne infections in two provinces of Northern Burkina Faso. The information on the DNA sequencing methodology was not indicated. Therefore, in my opinion the authors have to revise and correct the manuscript, especially material and methods and results section as well. Nonetheless, the overall merit and the results of this survey might be published in Pathogens. Please read and correct the following remarks.
Results:
[110] In the Table 1 the term 16-20 of the variable: No. of cattle examined does not represent any data. I suggest changing it to >16.
[132] You have missed two numbers in numeration of tables and figures. There is no table 2 and 3 in the text.
[133] The table 5 is unreadable. I suggest placing it horizontally on the page. Gene names the table should be written in italic. Moreover, the sequences were obtained not recovered.
[138] The authors frequently indicate that the almost complete 18S rRNA gene sequence was analysed, however nowhere in the text is indicated what part (in percent) of this sequence is actually analysed. Please add this information.
[149] You missed Figure 2 and 3.
[162] The font has changed to bigger one
[173] The font size has changed
Materials and Methods:
[324] Have the veterinarians examined animals taken to this study or the information is solely based on farmer’s information?
[370] There is a lack information about the methodology of sequencing itself. Was the Sanger sequencing performed or NGS? The sequences were read in capillary electrophoresis, on what device, what was the coverage of the sequences? Was it performed by external service or in the same laboratory as the rest of the analyses?
[388] If all sequences were registered in GenBank then you should indicate the number of BioProject assigned to this data.
[406] The font size has changed
Round 2
Reviewer 1 Report
The revised manuscript is much improved as a scientific writing